# ResMLP: Feedforward networks for image classification with data-efficient training

## Abstract

We present ResMLP, an architecture built entirely upon multi-layer perceptrons for image classification. It is a simple residual network that alternates (i) a linear layer in which image patches interact, independently and identically across channels, and (ii) a two-layer feed-forward network in which channels interact independently per patch. When trained with a modern training strategy using heavy data-augmentation and optionally distillation, it attains surprisingly good accuracy/complexity trade-offs on ImageNet. We also train ResMLP models in a self-supervised setup, to further remove priors from employing a labelled dataset. Finally, by adapting our model to machine translation we achieve surprisingly good results.

We will share our code based on the Timm library and pre-trained models.

## 1 Introduction

Recently, the transformer architecture [60], adapted from its original use in natural language processing with only minor changes, has achieved performance competitive with the state of the art on ImageNet-1k [50] when pre-trained with a sufficiently large amount of data [16]. Retrospectively, this achievement is yet another step towards less priors: convolutional neural networks had removed a lot of hand-made choices compared to hand-designed pre-CNN approaches, moving the paradigm of hard-wired features to hand-designed architectural choices. Vision transformers avoid making assumptions inherent to convolutional architectures and noticeably the translation invariance.

What these recent transformer-based works suggest is that longer training schedules, more parameters, more data [16] and/or more regularization [56], are sufficient to recover the important priors for tasks as complex as ImageNet classification. See also our discussion of related work in Section 4. This concurs with recent studies [2, 15] that better disentangle the benefits from the architectures from those of the training scheme.

In this paper, we push this trend further, and propose Residual Multi-Layer Perceptrons (ResMLP): a purely multi-layer perceptron (MLP) based architecture for image classification. We outline our architecture in Figure 1 and detail it further in Section 2. It is intended to be simple: it takes image patches as input, projects them with a linear layer, and sequentially updates them in turn with two residual operations: (i) a simple linear layer that provides interaction between the patches, which is applied to all channels independently; and (ii) an MLP with a single hidden layer, which is independently applied to all patches. At the end of the network, the patches are average pooled, and fed to a linear classifier.

This architecture is strongly inspired by the vision transformers (ViT) [16], yet it is much simpler in several ways: we do not use any form of attention, only linear layers along with the GELU non-linearity [25]. Since our architecture is much more stable to train than transformers, we do not

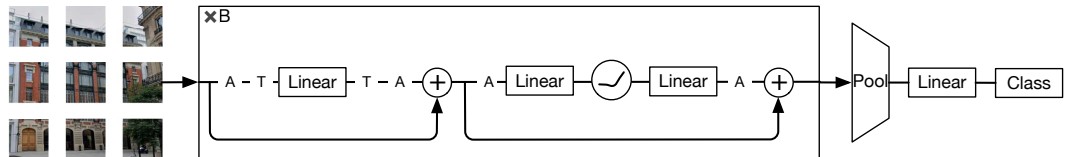

Figure 1: The ResMLP architecture: After linearly projecting the image patches, our network alternately processes them by (1) a communication layer between vectors implemented as a linear layer; (2) a two-layer residual perceptron. We denote by A the Affine element-wise transformation, and by T the transposition.

need batch-specific or cross-channel normalizations such as BatchNorm, GroupNorm or LayerNorm. Our training procedure mostly follows the one initially introduced for DeiT [56] and CaiT [57].

Due to its linear nature, the patch interactions in our model can be easily visualised and interpreted. While the interaction pattern learned in the first layer is very similar to a small convolutional filter, we observe more subtle interactions across patches in deeper layers. These include some form of axial filters, and long-range interactions early in the network.

In summary, in this paper, we show that

- despite their simplicity, Residual Multi-Layer Perceptrons reach surprisingly good accuracy/complexity trade-offs with ImageNet-1k training only[1], without requiring normalization based on batch or channel statistics;

- these models benefit significantly from distillation methods [56]; they are also compatible with modern self-supervised learning methods based on data augmentation, such as DINO [6];

- thank to its design where patch embeddings simply "communicate" through a linear layer, we can make observations on the spatial interaction that the network learns across layers;

- we adapt ResMLP to machine translation, and again obtain surprisingly good results.

## 2 Method

Our model, depicted in Figure 1, is inspired by the ViT model, from which we adopt the columnar structure with fixed-resolution feature maps. We proceed two drastic simplifications. We refer the reader to Dosovitskiy *et al.* [16] for more details about the ViT architecture.

**The overall ResMLP architecture.** Our model, denoted by ResMLP, takes a grid of $N \times N$ non-overlapping patches as input, where the patch size is typically equal to $16 \times 16$. The patches are then independently passed through a linear layer to form a set of $N^2$ $d$-dimensional embeddings.

The resulting set of $N^2$ embeddings are fed to a sequence of *Residual Multi-Layer Perceptron* layers to produce a set of $N^2$ $d$-dimensional output embeddings. These output embeddings are then averaged as a $d$-dimension vector to represent the image, which is fed to a linear classifier to predict the label associated with the image. Training uses the cross-entropy loss.

**The Residual Multi-Perceptron Layer.** Our network is a sequence of layers that all have the same structure: a linear sublayer followed by a feedforward sublayer. Similar to the Transformer layer, each sublayer is paralleled with a skip-connection [23]. We do not apply Layer Normalization [1] because training is stable without it when using the following Affine transformation:

$$\mathrm{Aff}_{\boldsymbol{\alpha},\boldsymbol{\beta}}(\mathbf{x}) = \mathrm{Diag}(\boldsymbol{\alpha})\mathbf{x} + \boldsymbol{\beta}, \tag{1}$$

where $\boldsymbol{\alpha}$ and $\boldsymbol{\beta}$ are learnable weight vectors. This operation simply rescales and shifts the input element-wise. Moreover, it has no cost at inference time, as it can absorbed in the adjacent linear layer. Note, when writing $\mathrm{Aff}(\mathbf{X})$ the operation is applied independently to each column of $\mathbf{X}$. While similar to BatchNorm [30] and Layer Normalization [1], the $\mathrm{Aff}$ operator does not depend on any batch statistics. Therefore, it is closer to the recent LayerScale method [57], which improves the

[1]Concurrent work by Tolstikhin *et al.* [55] brings complementary insights to ours: they achieve interesting performance with larger MLP models pre-trained on the larger public ImageNet-22k and even more data with the proprietary JFT-300M. In contrast, we focus on faster models trained on ImageNet-1k. Other concurrent related work includes that of Melas-Kyriazi [39] and the RepMLP [14] and gMLP [38] models.

optimization of deep transformers when initializing $\boldsymbol{\alpha}$ to a small value. Note, LayerScale does not have a bias term.

We apply this transformation twice for each residual block. As as a pre-normalization `Aff` replaces the LayerNormalization, and avoids using channel-wise statistics. Here, we initialize $\boldsymbol{\alpha} = \mathbf{1}$, and $\boldsymbol{\beta} = \mathbf{0}$. As a post-processing of the residual block, `Aff` implements LayerScale and therefore we follow the same small value initialization for $\boldsymbol{\alpha}$ as in [57] for the post-normalization.

Finally, we follow the same structure for the feedforward sublayer as in the Transformer; we only replace the `ReLU` non-linearity by a `GELU` function [25].

Overall, our Multi-layer perceptron takes a set of $N^2$ $d$-dimensional input features stacked in a $d \times N^2$ matrix $\mathbf{X}$, and outputs a set of $N^2$ $d$-dimension output features, stacked in a matrix $\mathbf{Y}$ with the following set of transformations:

$$\mathbf{Z} = \mathbf{X} + \texttt{Aff}\left(\left(\mathbf{A}\,\texttt{Aff}\left(\mathbf{X}\right)^{\top}\right)^{\top}\right), \tag{2}$$

$$\mathbf{Y} = \mathbf{Z} + \texttt{Aff}\left(\mathbf{C}\,\texttt{GELU}(\mathbf{B}\,\texttt{Aff}(\mathbf{Z}))\right), \tag{3}$$

where $\mathbf{A}$, $\mathbf{B}$ and $\mathbf{C}$ are the main learnable weight matrices of the layer. The dimensions of the parameter matrix $\mathbf{A}$ are $N^2 \times N^2$, *i.e.*, this sublayer exchanges information across all the locations, while the feedforward sublayer works per location. As a consequence, the intermediate activation matrix $\mathbf{Z}$ has the same dimensions as the matrices $\mathbf{X}$ and $\mathbf{Y}$. Finally, the weight matrices $\mathbf{B}$ and $\mathbf{C}$ have the same dimensions as in a Transformer layer, which are $4d \times d$ and $d \times 4d$, respectively.

The main difference compared to a Transformer layer is that we replace the self-attention by the linear interaction defined in Eq. (2). While self-attention computes a convex combination of other features with coefficients that are data dependent, the linear interaction layer in Eq. (2) computes a general linear combination using learned coefficients that are not data dependent. As compared to a convolutional layers which have local support and share weights across space, our linear patch interaction layer offers a global support and does not share weights, moreover it is applied independently across channels.

**Relationship to the Vision Transformer.** Our model can be regarded as a drastic simplification of the ViT model by Dosovitskiy *et al.* [16]. We depart from this model as follows:

- We do not include any self-attention block. Instead we have a linear patch interaction layer without any non-linearity.

- We do not have the extra "class" token that is typically used in these models to aggregate information via attention. Instead, we simply use average pooling. We do, however, also consider a specific aggregation layer as a variant, which we describe in the next paragraph.

- We do not include any form of positional embedding: the linear communication module between patches implicitly takes into account the patch position.

- Instead of pre-LayerNormalization, we use a simple learnable affine transform, thus avoiding any form of batch and channel-wise statistics.

**Class-MLP.** As an alternative to average pooling, we also experimented with an adaptation of the class-attention introduced in CaiT [57]. In CaiT, this consists of two layers that have the same structure as the transformer, but in which only the class token is updated based on the frozen patch embeddings. We translate this method to our architecture, except that, after aggregating the patches with a linear layer, we replace the attention-based interaction between the class and patch embeddings by simple linear layers, still keeping the patch embeddings frozen. This increases the performance, at the expense of adding some parameters and computational cost. We refer to this pooling variant as "class-MLP", since the purpose of these few layers is to replace average pooling.

## 3 Experiments

In this section, we present experimental results for our ResMLP architecture for image classification. We also study the impact of the different components in a series of ablations. We consider three training paradigms in our experiments:

Table 1: **Comparison between architectures on ImageNet classification.** We compare different architectures based on convolutional networks, Transformers and feedforward networks with comparable FLOPs and number of parameters. We report Top-1 accuracy on the validation set of ImageNet-1k with different measure of complexity: throughput, FLOPs, number of parameters and peak memory usage. All the models use $224 \times 224$ images as input. By default the Transformers and feedforward networks uses $14 \times 14$ patches of size $16 \times 16$, see Table 3 for the detailed specification of our main models. The throughput is measured on a single V100-32GB GPU with batch size fixed to 32. For reference, we include the state of the art with ImageNet training only.

| | Arch. | #params ($\times 10^6$) | throughput (im/s) | FLOPS ($\times 10^9$) | Peak Mem (MB) | Top-1 Acc. |
|---|---|---|---|---|---|---|
| *State of the art* | CaiT-M48↑448Υ [57] | 356 | 5.4 | 329.6 | 5477.8 | 86.5 |
| | NfNet-F6 SAM [5] | 438 | 16.0 | 377.3 | 5519.3 | 86.5 |
| *Convolutional networks* | EfficientNet-B3 [53] | 12 | 661.8 | 1.8 | 1174.0 | 81.1 |
| | EfficientNet-B4 [53] | 19 | 349.4 | 4.2 | 1898.9 | 82.6 |
| | EfficientNet-B5 [53] | 30 | 169.1 | 9.9 | 2734.9 | 83.3 |
| | RegNetY-4GF [47] | 21 | 861.0 | 4.0 | 568.4 | 80.0 |
| | RegNetY-8GF [47] | 39 | 534.4 | 8.0 | 841.6 | 81.7 |
| | RegNetY-16GF [47] | 84 | 334.7 | 16.0 | 1329.6 | 82.9 |
| *Transformer networks* | DeiT-S [56] | 22 | 940.4 | 4.6 | 217.2 | 79.8 |
| | DeiT-B [56] | 86 | 292.3 | 17.5 | 573.7 | 81.8 |
| | CaiT-XS24 [57] | 27 | 447.6 | 5.4 | 245.5 | 81.8 |
| *Feedforward networks* | ResMLP-S12 | 15 | 1415.1 | 3.0 | 179.5 | 76.6 |
| | ResMLP-S24 | 30 | 715.4 | 6.0 | 235.3 | 79.4 |
| | ResMLP-B24 | 116 | 231.3 | 23.0 | 663.0 | 81.0 |

- *Supervised learning:* We train ResMLP from labeled images with a softmax classifier and cross-entropy loss. This paradigm is the main focus of our work.

- *Self-supervised learning:* We train the ResMLP architecture without labels. We consider the DINO method of Caron *et al.* [6] that trains a network by distilling knowledge from previous instances of the same network, leading to a form of self-distillation without labels.

- *Knowledge distillation:* We employ the knowledge distillation procedure proposed by Touvron *et al.* [56] to guide the supervised training of ResMLP with a convnet.

## 3.1 Experimental setting

**Datasets.** We train our models on the ImageNet-1k dataset [50], that contains 1.2M images evenly spread over 1,000 object categories. In the absence of an available test set for this benchmark, we follow the standard practice in the community by reporting performance on the validation set. This is not ideal since the validation set was originally designed to select hyper-parameters. Comparing methods on this set may not be conclusive enough because an improvement in performance may not be caused by better modeling, but by a better selection of hyper-parameters. To mitigate this risk, we report additional results in transfer learning and on two alternative versions of ImageNet that have been built to have distinct validation and test sets, namely the ImageNet-real [3] and ImageNet-v2 [49] datasets. We also report a few data-points when training on ImageNet-21k. Our hyper-parameters are mostly adopted from Touvron et al. [56, 57].

**Hyper-parameter settings.** In the case of supervised learning, we train our network with the Lamb optimizer [63] with a learning rate of $5 \times 10^{-3}$ and weight decay 0.2. We initialize the LayerScale parameters as a function of the depth by following CaiT [57]. The rest of the hyper-parameters follow the default setting used in DeiT [56]. For the knowledge distillation paradigm, we use the same RegNety-16GF [48] as in DeiT with the same training schedule. The majority of our models take two days to train on eight V100-32GB GPUs.

## 3.2 Main Results

In this section, we compare our architecture to models with more conventional network architectures of comparable size and throughput on ImageNet.

**Comparison with Transformers and convnets in a supervised setting.** In Table 1, we compare ResMLP with different convolutional and Transformer architectures. For completeness, we report

Table 2: **Self-supervised learning** with DINO [6]. Classification accuracy on ImageNet-1k val. ResMLPs evaluated with linear and $k$-NN evaluation on ImageNet are comparable to convnets but inferior to ViT.

| Models | ResNet-50 | ViT-S/16 | ViT-S/8 | ViT-B/16 | ResMLP-S12 | ResMLP-S24 |
|---|---|---|---|---|---|---|
| Params. ($\times 10^6$) | 25 | 22 | 22 | 87 | 15 | 30 |
| FLOPS ($\times 10^9$) | 4.1 | 4.6 | 22.4 | 17.5 | 3.0 | 6.0 |
| Linear | 75.3 | 77.0 | 79.7 | 78.2 | 67.5 | 72.8 |
| $k$-NN | 67.5 | 74.5 | 78.3 | 76.1 | 62.6 | 69.4 |

the best-published numbers obtained with a model trained on ImageNet alone. As expected, in terms of the trade-off between accuracy, FLOPs, and throughput, ResMLP is not as efficient as convolutional networks or Transformers. However, their accuracy is encouraging: we compare them with architectures that have benefited from years of research and careful optimization towards these trade-offs. Overall, our results suggest that the structural constraints imposed by the layer design do not have a drastic influence on performance, especially when training models with enough data and recent advances in training and regularization.

**Self-supervised pre-training of ResMLP.** We explore the possibility of training ResMLP using DINO, a recent self-supervised learning approach [6]. We pre-train ResMLP-S12 models with this approach during 300 epochs. We report our results in Table 2. As expected given the supervised classification results, the accuracies obtained with ResMLP are less good than with ViT. Nevertheless, the performance is surprisingly high for a pure MLP architecture and competitive with Convnet in knn evaluation. We hope that these result will serve as a baseline for future work.

After self-supervised pre-training, we also fine-tune the network on ImageNet using ground truth labels. This pre-training substantially improves the accuracy, when comparing with the same model ResMLP-S24 solely trained with labels (top-1 acc. of 79.9% on ImageNet-val instead of 79.4% for ResMLP-S24, with the same total number of epochs). Results on ImageNet-v2 suggest that it reduces overfitting (68.6% on ImageNet-v2, vs 67.9% with supervised training only).

**Improving models with knowledge distillation.** We study our model when training following the knowledge distillation approach of Touvron *et al.* [56]. In their work, the authors show the impact of training a ViT model by distilling it from a RegNet. In this experiment, we explore if ResMLP also benefits from this procedure and summarize our results in Table 3 (Blocks "Baseline models" and "Training"). We observe that similar to DeiT models, ResMLP greatly benefits from distilling from a convnet. This result concurs with the observations made by d'Ascoli *et al.* [13], who used convnets to initialize feedforward networks. Even though our setting differs from theirs in scale, the problem of overfitting for feedforward networks is still present on ImageNet. The additional regularization obtained from the distillation is a possible explanation for this improvement.

**Visualisation.** Because they are linear, our patch interaction layers from Eq. (2) are easily interpretable. In Figure 2 we visualise the rows of the interaction matrices **A** as $N \times N$ images, for our ResMLP-S24 model. The early layers show convolution-like patterns: the learned weights resemble shifted versions of each other and have local support. Interestingly, in many layers, the support also extends along both axes, most prominently seen in layer seven. The last seven layers of the network are different: they consist of a spike for the patch itself and a diffuse response across other patches with larger or smaller magnitude; see layer 20.

### 3.3 Visualization & analysis of the linear interaction between patches

**Measuring sparsity of the weights.** The visualizations described above suggest that the linear communication layers are sparse. We analyze this quantitatively in more detail in Figure 3. We measure the sparsity of the matrix **A**, and compare it to the sparsity of **B** and **C** from the per-patch MLP. Since there are no exact zeros, we measure the rate of components whose absolute value is lower than 5% of the maximum value. Note, discarding the small values is analogous to the case where we normalize the matrix by its maximum and use a finite-precision representation of weights. For instance, with a 4-bits representation of weight, one would typically round to zero all weights whose absolute value is below 6.25% of the maximum value.

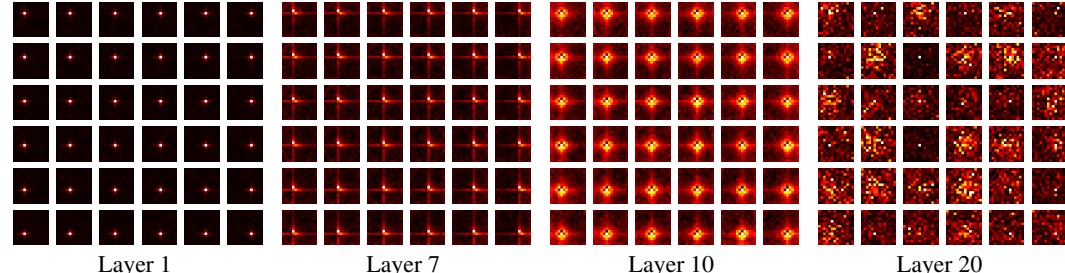

Layer 1       Layer 7       Layer 10       Layer 20

Figure 2: **Visualisation of the linear layers in ResMLP-S24.** For each layer we visualise the rows of the matrix **A** as a set of $14 \times 14$ pixel images, for sake of space we only show the rows corresponding to the $6 \times 6$ central patches. We observe patterns in the linear layers that share similarities with convolutions. In appendix B we provide comparable visualizations for all layers of a ResMLP-S12 model.

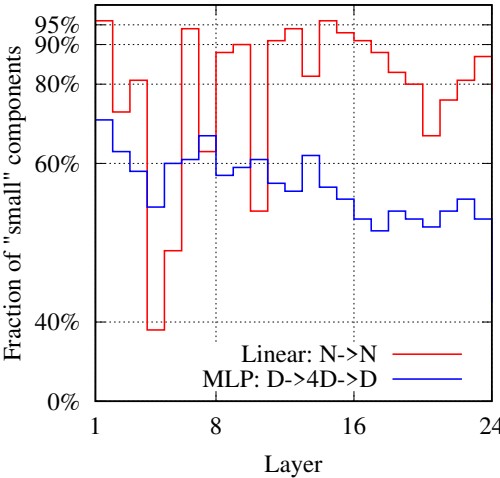

Figure 3: **Sparsity of linear interaction layers.** For each layer (linear and MLP), we show the rate of components whose absolute value is lower than 5% of the maximum. Linear interaction layers are sparser than the matrices involved in the per-patch MLP.

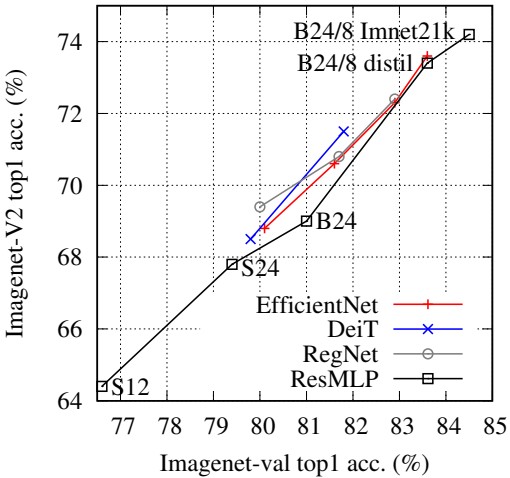

Figure 4: **Top-1 accuracy on ImageNet-V2 *vs*. ImageNet-val.** ResMLPs tend to overfit slightly more under identical training method. This is partially alleviated with by introducing more regularization (more data or distillation, see e.g., ResMLP-B24/8-distil).

The measurements in Figure 3 show that all three matrices are sparse, with the layers implementing the patch communication being significantly more so. This suggests that they may be compatible with parameter pruning, or better, with modern quantization techniques that induce sparsity at training time, such as Quant-Noise [20] and DiffQ [19]. The sparsity structure, in particular in earlier layers, see Figure. 2, hints that we could implement the patch interaction linear layer with a convolution. We provide some results for convolutional variants in our ablation study. Further research on network compression is beyond the scope of this paper, yet we believe it worth investigating in the future.

**Communication across patches** if we remove the linear interaction layer (linear $\rightarrow$ none), we obtain substantially lower accuracy (-20% top-1 acc.) for a "bag-of-patches" approach. We have tried several alternatives for the linear patch interaction layer, which are presented in Table 3 (block "patch communication"). Amongst them, using the same MLP structure as for patch processing (linear $\rightarrow$ MLP), which we analyze in more details in the supplementary material. The simpler choice of a single linear square layer led to a better accuracy/performance trade-off – considering that the MLP variant requires compute halfway between ResMLP-S12 and ResMLP-S24 – and requires fewer parameters than a residual MLP block.

The visualization in Figure 2 indicates that many linear interaction layers look like convolutions. In our ablation, we replaced the linear layer with different types of $3 \times 3$ convolutions. The depth-wise convolution does not implement interaction across channels – as our linear patch communication layer – and yields similar performance at a comparable number of parameters and FLOPs. While full $3 \times 3$ convolutions yield best results, they come with roughly double the number of parameters

| Ablation | Model | Patch size | Params ×10^6 | FLOPs ×10^9 | Variant | top-1 acc. on ImageNet | | |
|---|---|---|---|---|---|---|---|---|
| | | | | | | val | real [3] | v2 [49] |
| Baseline models | ResMLP-S12 | 16 | 15.4 | 3.0 | 12 layers, working dimension 384 | 76.6 | 83.3 | 64.4 |
| | ResMLP-S24 | 16 | 30.0 | 6.0 | 24 layers, working dimension 384 | 79.4 | 85.3 | 67.9 |
| | ResMLP-B24 | 16 | 115.7 | 23.0 | 24 layers, working dimension 768 | 81.0 | 86.1 | 69.0 |
| Normalization | ResMLP-S12 | 16 | 15.4 | 3.0 | Aff → Layernorm | 77.7 | 84.1 | 65.7 |
| Pooling | ResMLP-S12 | 16 | 17.7 | 3.0 | average pooling → Class-MLP | 77.5 | 84.0 | 66.1 |
| Patch communication | ResMLP-S12 | 16 | 14.9 | 2.8 | linear → none | 56.5 | 63.4 | 43.1 |
| | ResMLP-S12 | 16 | 18.6 | 4.3 | linear → MLP | 77.3 | 84.0 | 65.7 |
| | ResMLP-S12 | 16 | 30.8 | 6.0 | linear → conv 3x3 | 77.3 | 84.4 | 65.7 |
| | ResMLP-S12 | 16 | 14.9 | 2.8 | linear → conv 3x3 depth-wise | 76.3 | 83.4 | 64.6 |
| | ResMLP-S12 | 16 | 16.7 | 3.2 | linear → conv 3x3 depth-separable | 77.0 | 84.0 | 65.5 |
| Patch size | ResMLP-S12/14 | 14 | 15.6 | 4.0 | patch size 16×16→14×14 | 76.9 | 83.7 | 65.0 |
| | ResMLP-S12/8 | 8 | 22.1 | 14.0 | patch size 16×16→8×8 | 79.1 | 85.2 | 67.2 |
| | ResMLP-B24/8 | 8 | 129.1 | 100.2 | patch size 16×16→8×8 | 81.0 | 85.7 | 68.6 |
| Training | ResMLP-S12 | 16 | 15.4 | 3.0 | old-fashioned (90 epochs) | 69.2 | 76.0 | 56.1 |
| | ResMLP-S12 | 16 | 15.4 | 3.0 | pre-trained SSL (DINO) | 76.5 | 83.6 | 64.5 |
| | ResMLP-S12 | 16 | 15.4 | 3.0 | distillation | 77.8 | 84.6 | 66.0 |
| | ResMLP-S24 | 16 | 30.0 | 6.0 | pre-trained SSL (DINO) | 79.9 | 85.9 | 68.6 |
| | ResMLP-S24 | 16 | 30.0 | 6.0 | distillation | 80.8 | 86.6 | 69.8 |
| | ResMLP-B24/8 | 8 | 129.1 | 100.2 | distillation | 83.6 | 88.4 | 73.4 |
| | ResMLP-B24/8 | 8 | 129.1 | 100.2 | pre-trained ImageNet-21k (60 epochs) | 84.4 | 88.9 | 74.2 |

Table 3: **Ablation.** Our default configurations are presented in the three first rows. By default we train during 400 epochs. The "old-fashioned" is similar to what was employed for ResNet [23]: SGD, 90-epochs waterfall schedule, same augmentations up to variations due to library used.

and FLOPs. Interestingly, the depth-separable convolutions combine accuracy close to that of full $3\times3$ convolutions with a number of parameters and FLOPs comparable to our linear layer. This suggests that convolutions on low-resolution feature maps at all layers is an interesting alternative to the common pyramidal design of convnets, where early layers operate at higher resolution and smaller feature dimension.

### 3.4 Ablation studies

Table 3 reports the ablation study of our base network and a summary of our preliminary exploratory studies. We discuss the ablation below and give more detail about early experiments in Appendix A.

**Control of overfitting.** Since MLPs are subject to overfitting, we show in Fig. 4 a control experiment to probe for problems with generalization. We explicitly analyze the differential of performance between the ImageNet-val and the distinct ImageNet-V2 test set. The relative offsets between curves reflect to which extent models are overfitted to ImageNet-val w.r.t. hyper-parameter selection. The degree of overfitting of our MLP-based model is overall neutral or slightly higher to that of other transformer-based architectures or convnets with same training procedure.

**Normalization & activation.** Our network configuration does not contain any batch normalizations. Instead, we use the affine per-channel transform Aff. This is akin to Layer Normalization [1], typically used in transformers, except that we avoid to collect any sort of statistics, since we do no need it it for convergence. In preliminary experiments with pre-norm and post-norm [24], we observed that both choices converged. Pre-normalization in conjunction with Batch Normalization could provide an accuracy gain in some cases, see Appendix A.

We choose to use a GELU [25] function. In Appendix A we also analyze the activation function: ReLU [22] also gives a good performance, but it was a bit more unstable in some settings. We did not manage to get good results with SiLU [25] and HardSwish [28].

**Pooling.** Replacing average pooling with Class-MLP, see Section 2, brings a significant gain for a negligible computational cost. We do not include it by default to keep our models more simple.

**Patch size.** Smaller patches significantly increase the performance, but also increase the number of flops (see Block "Patch size" in Table 3). Smaller patches benefit more to larger models, but only with an improved optimization scheme involving more regularization (distillation) or more data.

**Training.** Consider the Block "Training' in Table 3. ResMLP significantly benefits from modern training procedures such as those used in DeiT. For instance, the DeiT training procedure improves

| Architecture | FLOPs | Res. | CIFAR$_{10}$ | CIFAR$_{100}$ | Flowers102 | Cars | iNat$_{18}$ | iNat$_{19}$ |
|---|---|---|---|---|---|---|---|---|
| EfficientNet-B7 [53] | 37.0B | 600 | 98.9 | 91.7 | 98.8 | 94.7 | _ | _ |
| ViT-B/16 [16] | 55.5B | 384 | 98.1 | 87.1 | 89.5 | _ | _ | _ |
| ViT-L/16 [16] | 190.7B | 384 | 97.9 | 86.4 | 89.7 | _ | _ | _ |
| Deit-B/16 [56] | 17.5B | 224 | 99.1 | 90.8 | 98.4 | 92.1 | 73.2 | 77.7 |
| ResNet50 [58] | 4.1B | 224 | _ | _ | 96.2 | 90.0 | 68.4 | 73.7 |
| Grafit/ResNet50 [58] | 4.1B | 224 | _ | _ | 97.6 | 92.7 | 68.5 | 74.6 |
| ResMLP-S12 | 3.0B | 224 | 98.1 | 87.0 | 97.4 | 84.6 | 60.2 | 71.0 |
| ResMLP-S24 | 6.0B | 224 | 98.7 | 89.5 | 97.9 | 89.5 | 64.3 | 72.5 |

Table 4: **Evaluation on transfer learning.** Classification accuracy (top-1) of models trained on ImageNet-1k for transfer to datasets covering different domains. The ResMLP architecture takes $224 \times 224$ images during training and transfer, while ViTs and EfficientNet-B7 work with higher resolutions, see "Res." column.

the performance of ResMLP-S12 by $7.4\%$ compared to the training employed for ResNet [23][2]. This is in line with recent work pointing out the importance of the training strategy over the model choice [2, 48]. Pre-training on more data and distillation also improve the performance of ResMLP, especially for the bigger models, *e.g.*, distillation improves the accuracy of ResMLP-B24/8 by $2.6\%$.

**Other analysis.** In our early exploration, we evaluated several alternative design choices. As in transformers, we could use positional embeddings mixed with the input patches. In our experiments we did not see any benefit from using these features, see Appendix A. This observation suggests that our linear patch interaction layer provides sufficient spatial communication, and referencing absolute positions obviates the need for any form of positional encoding.

## 3.5 Transfer learning

We evaluate the quality of features obtained from a ResMLP architecture when transferring them to other domains. The goal is to assess if the features generated from a feedforward network are more prone to overfitting on the training data distribution. We adopt the typical setting where we pre-train a model on ImageNet-1k and fine-tune it on the training set associated with a specific domain. We report the performance with different architectures on various image benchmarks in Table 4, namely CIFAR-10 and CIFAR-100 [34], Flowers-102 [42], Stanford Cars [33] and iNaturalist [27]. We refer the reader to the corresponding references for a more detailed description of the datasets. We observe that the performance of our ResMLP is competitive with the existing architectures, showing that pretraining feedforward models with enough data and regularization via data augmentation greatly reduces their tendency to overfit on the original distribution. Interestingly, this regularization also prevents them from overfitting on the training set of smaller dataset during the fine-tuning stage.

## 3.6 Machine translation

We also evaluate the ResMLP transpose-mechanism to replace the self-attention in the encoder and decoder of a neural machine translation system. We train models on the WMT 2014 English-German and English-French tasks, following the setup from Ott *et al.* [45]. We consider models of dimension 512, with a hidden MLP size of 2048, and with 6 or 12 layers. Note that the current state of the art employs much larger models: our 6 layers model is more comparable to the base transformer model from Vaswani *et al.* [60], which serves as a baseline, along with pre-transformer architectures such as Recurrent and convolutional neural networks. We use Adagrad with learning rate 0.2, 32k steps of linear warmup, label smoothing 0.1, dropout rate 0.15 for en-de and 0.1 for en-fr. We initialize the LayerScale parameter to 0.2. We generate translations with the beam search algorithm, with a beam of size 4. As shown in Table 5, the results are at least on par with other architectures:

Table 5: **Machine translation** on WMT 2014 translation tasks. We report tokenized BLEU on *newstest2014*.

| Models | GNMT [61] | ConvS2S [21] | Transf. (base) [60] | ResMLP-6 | ResMLP-12 |
|---|---|---|---|---|---|
| EN-DE | 24.6 | 25.2 | 27.3 | 26.4 | 26.8 |
| EN-FR | 39.9 | 40.5 | 38.1 | 40.3 | 40.6 |

---

[2]Interestingly, if trained with this "old-fashion" setting, ResMLP-S12 outperforms AlexNet [35] by a margin.

## 4 Related work

We review the research on applying Fully Connected Network (FCN) for computer vision problems as well as other architectures that shares common modules with our model.

**Fully-connected network for images.** Many studies have shown that FCNs are competitive with convnets for the tasks of digit recognition [11, 51], keyword spotting [7] and handwritting recognition [4]. Several works [37, 40, 59] have questioned if FCNs are also competitive on natural image datasets, such as CIFAR-10 [34]. More recently, d'Ascoli *et al.* [13] have shown that a FCN initialized with the weights of a pretrained convnet achieves performance that are superior than the original convnet. Neyshabur [41] further extend this line of work by achieving competitive performance by training an FCN from scratch but with a regularizer that constrains the models to be close to a convnet. These studies have been conducted on small scale datasets with the purpose of studying the impact of architectures on generalization in terms of sample complexity [18] and energy landscape [31]. In our work, we show that, in the larger scale setting of ImageNet, FCNs can attain surprising accuracy without any constraint or initialization inspired by convnets.

Finally, the application of FCN networks in computer vision have also emerged in the study of the properties of networks with infinite width [43], or for inverse scattering problems [32]. More interestingly, the Tensorizing Network [44] is an approximation of very large FCN that shares similarity with our model, in that they intend to remove prior by approximating even more general tensor operations, *i.e.*, not arbitrarily marginalized along some pre-defined sharing dimensions. However, their method is designed to compress the MLP layers of a standard convnets.

**Other architectures with similar components.** Our FCN architecture shares several components with other architectures, such as convnets [35, 36] or transformers [60]. A fully connected layer is equivalent to a convolution layer with a $1 \times 1$ receptive field, and several work have explored convnet architectures with small receptive fields. For instance, the VGG model [52] uses $3 \times 3$ convolutions, and later, other architectures such as the ResNext [62] or the Xception [10] mix $1 \times 1$ and $3 \times 3$ convolutions. In contrast to convnets, in our model interaction between patches is obtained via a linear layer that is shared across channels, and that relies on absolute rather than relative positions.

More recently, transformers have emerged as a promising architecture for computer vision [9, 17, 46, 56, 66]. In particular, our architecture takes inspiration from the structure used in the Vision Transformer (ViT) [17], and as consequence, shares many components. Our model takes a set of non-overlapping patches as input and passes them through a series of MLP layers that share the same structure as ViT, replacing the self-attention layer with a linear patch interaction layer. Both layers have a global field-of-view, unlike convolutional layers. Whereas in self-attention the weights to aggregate information from other patches are data dependent through queries and keys, in ResMLP the weights are not data dependent and only based on absolute positions of patches. In our implementation we follow the improvements of DeiT [56] to train vision transformers, use the skip-connections from ResNets [23] with pre-normalization of the layers [8, 24].

Finally, our work questions the importance of self-attention in existing architectures. Similar observations have been made in natural language processing. Notably, Synthesizer [54] shows that dot-product self-attention can be replaced by a feedforward network, with competitive performance on sentence representation benchmarks. As opposed to our work, Synthesizer does use data dependent weights, but in contrast to transformers the weights are determined from the queries only.

## 5 Conclusion

In this paper we have shown that a simple residual architecture, whose residual blocks consist of a one-hidden layer feed-forward network and a linear patch interaction layer, achieves an unexpectedly high performance on ImageNet classification benchmarks, provided that we adopt a modern training strategy such as those recently introduced for transformer-based architectures. Thanks to their simple structure, with linear layers as the main mean of communication between patches, we can vizualize the filters learned by this simple MLP. While some of the layers are similar to convolutional filters, we also observe sparse long-range interactions as early as the second layer of the network. We hope that our model free of spatial priors will contribute to further understanding of what networks with less priors learn, and potentially guide the design choices of future networks without the pyramidal design prior adopted by most convolutional neural networks.

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
