# ResMLP: Feedforward networks for image classification with data-efficient training

## Appendix

## A    Report on our exploration phase

As discussed in the main paper, our work on designing a residual multi-layer perceptron was inspired by the Vision Transformer. For our exploration, we have adopted the recent CaiT variant [57] as a starting point. This transformer-based architecture achieves state-of performance with Imagenet-training only (achieving 86.5% top-1 accuracy on Imagenet-val for the best model). Most importantly, the training is relatively stable with increasing depth.

In our exploration phase, our objective was to radically simplify this model. For this purpose, we have considered the Cait-S24 model for faster iterations. This network consists of 24-layer with a working dimension of 384. All our experiments below were carried out with images in resolution $224 \times 224$ and $N = 16 \times 16$ patches. Trained with regular supervision, Cait-S24 attains 82.7% top-1 acc. on Imagenet.

**SA $\rightarrow$ MLP.**    The self-attention can be seen a weight generator for a linear transformation on the values. Therefore, our first design modification was to get rid of the self-attention by replacing it by a residual feed-forward network, which takes as input the *transposed* set of patches instead of the patches. In other terms, in this case we alternate residual blocks operating along the channel dimension with some operating along the patch dimension. In that case, the MLP replacing the self-attention consists of the sequence of operations

$$(\cdot)^T \text{ --- linear } N \times 4N \text{ --- GELU --- linear } 4N \times N \text{ --- } (\cdot)^T$$

Hence this network is symmetrical in $N$ and $d$. By keeping the other elements identical to CaiT, the accuracy drops to $80.2\%$ (-2.5%) when replacing self-attention layers.

**Class-attention $\rightarrow$ class-MLP.**    If we further replace the class-attention layer of CaiT by a MLP as described in our paper, then we obtain an attention-free network whose top-1 accuracy on Imagenet-val is 79.2%, which is comparable to a ResNet-50 trained with a modern training strategy. This network has served as our baseline for subsequent ablations. Note that, at this stage, we still include LayerScale, a class embedding (in the class-MLP stage) and positional encodings.

**Distillation.**    The same model trained with distillation inspired by Touvron et al. [56] achieves 81.5%. The distillation variant we choose corresponds to the "hard-distillation", whose main advantage is that it does not require any parameter-tuning compared to vanilla cross-entropy. Note that, in all our experiments, this distillation method seems to bring a gain that is complementary and seemingly almost orthogonal to other modifications.

**Activation: LayerNorm $\rightarrow$ X.**    We have tried different activations on top of the aforementioned MLP-based baseline, and kept GeLU for its accuracy and to be consistent with the transformer choice.

| Activation | top-1 acc. |
|---|---|
| GeLU (baseline) | 79.2% |
| SILU | 78.7% |
| Hard Swish | 78.8% |
| ReLU | 79.1% |

**Ablation on the size of the communication MLP.** For the MLP that replaced the class-attention, we have explored different sizes of the latent layer, by adjusting the expansion factor $e$ in the sequence: linear $N \times e \times N$ — GELU — linear $e \times N \times N$. For this experiment we used average pooling to aggregating the patches before the classification layer.

| expansion factor $\times e$ | $\times 0.25$ | $\times 0.5$ | $\times 1$ | $\times 2$ | $\times 3$ | $\times 4$ |
|---|---|---|---|---|---|---|
| Imnet-val top-1 acc. | 78.6 | 79.2 | 79.2 | 79.3 | 78.8 | 78.8 |

We observe that a large expansion factor is detrimental in the patch communication, possibly because we should not introduce too much capacity in this residual block. This has motivated the choice of adopting a simple linear layer of size $N \times N$: This subsequently improved performance to 79.5% in a setting comparable to the table above. Additionally, as shown earlier this choice allows visualizations of the interaction between patches.

**Normalization.** On top of our MLP baseline, we have tested different variations for normalization layers. We report the variation in performance below.

| Pre-normalization | top-1 acc. |
|---|---|
| Layernorm (baseline) | 79.2% |
| Batch-Norm | +0.8% |
| $\ell_2$-norm | +0.4% |
| no norm (`Aff`) | +0.4% |

For the sake of simplicity, we therefore adopted only the `Aff` transformation so as to not depend on any batch or channel statistics.

**Position encoding.** In our experiments, removing the position encoding does not change the results when using a MLP or a simple linear layer as a communication mean across patch embeddings. This is not surprising considering that the linear layer implicitly encodes each patch identity as one of the dimension, and that additionally the linear includes a bias that makes it possible to differentiate the patch positions before the shared linear layer.

# B  Analysis of interaction layers in 12-layer networks

In this section we further analyze the linear interaction layers in 12-layer models.

In Figure B.1 we consider a ResMLP-S12 model trained on the ImageNet-1k dataset, as explained in Section 3.1, and show all the 12 linear patch interaction layers. The linear interaction layers in the supervised 12-layer model are similar to those observed in the 24-layer model in Figure 2.

We also provide the corresponding sparsity measurements for this model in Figure B.2, analogous to the measurements in Figure 3 for the supervised 24-layer model. The sparsity levels in the supervised 12-layer model (left panel) are similar to those observes in the supervised 24-layer model, cf. Figure 3. In the right panel of Figure B.2 we consider the sparsity levels of the Distilled 12-layer model, which are overall similar to those observed for supervised the 12-layer and 24-layer models.

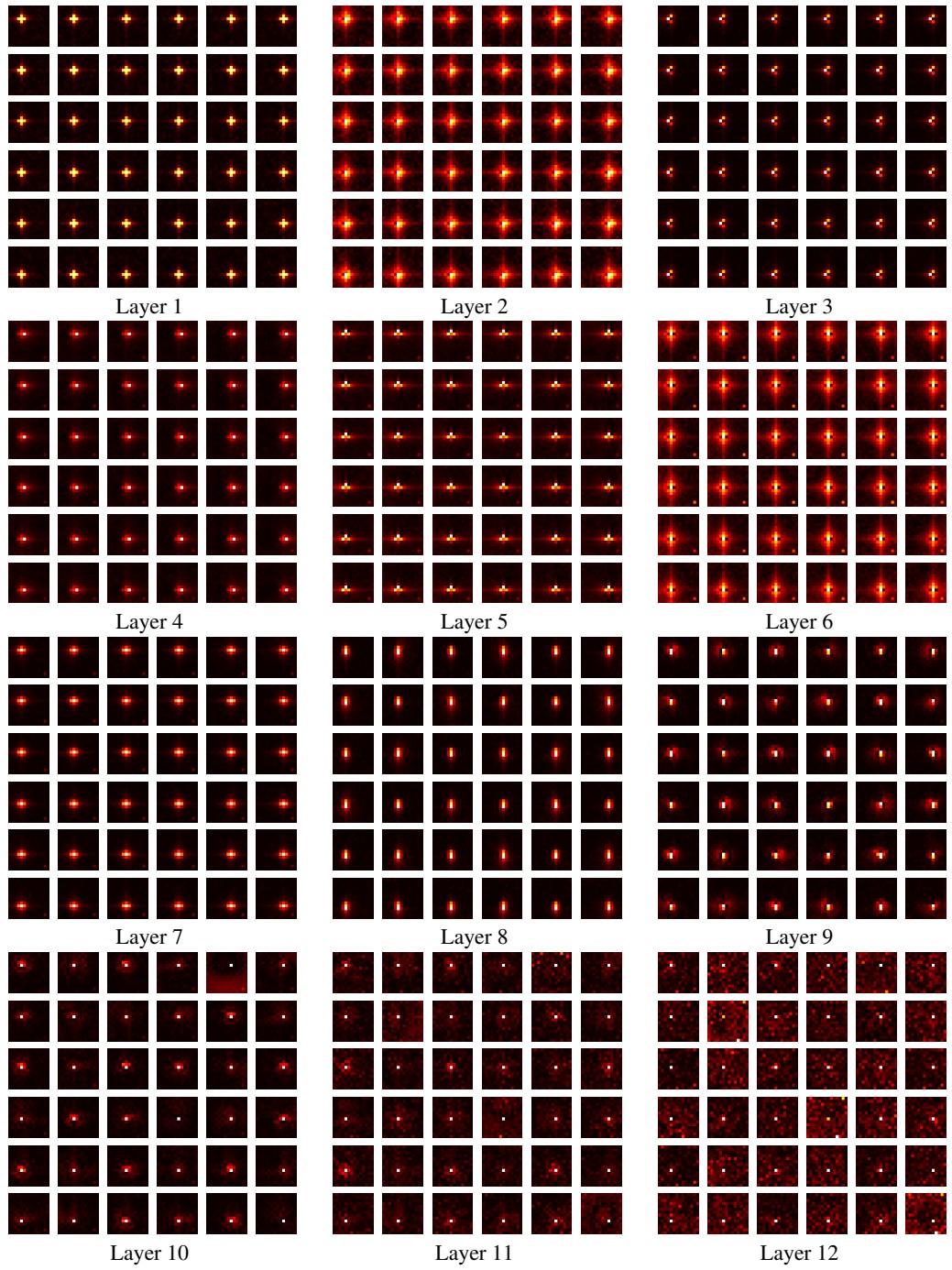

Figure B.1: **Visualisation of the linear interaction layers in the supervised ResMLP-S12 model.** For each layer we visualise the rows of the matrix $\mathbf{A}$ as a set of $14 \times 14$ pixel images, for sake of space we only show the rows corresponding to the $6{\times}6$ central patches.

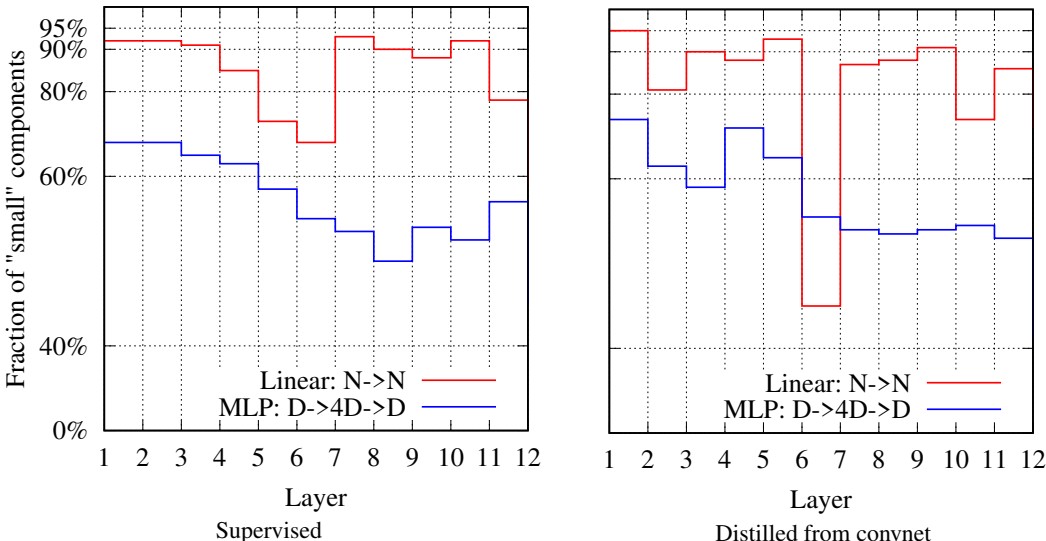

Figure B.2: Degree of sparsity (fraction of values small than 5% of the maximum) for Linear and MLP layers, for ResMLP-S12 networks. The network trained in supervised mode and the one learned with distillation overall have a comparable degree of sparsity. The self-supervised model, trained during 300 epochs *vs.* 400 for the other ones, is less sparse on the patch communication linear layer.

## C   Model definition in Pytorch

In Algorithm 1 we provide the pseudo-pytorch-code associated with our model.

---

**Algorithm 1** Pseudocode of ResMLP in PyTorch-like style

---

```python
# No norm layer
class Affine(nn.Module):
    def __init__(self, dim):
        super().__init__()
        self.alpha = nn.Parameter(torch.ones(dim))
        self.beta = nn.Parameter(torch.zeros(dim))
    def forward(self, x):
        return self.alpha * x + self.beta

# MLP on channels
class Mlp(nn.Module):
    def __init__(self, dim):
        super().__init__()
        self.fc1 = nn.Linear(dim, 4 * dim)
        self.act = nn.GELU()
        self.fc2 = nn.Linear(4 * dim, dim)
    def forward(self, x):
        x = self.fc1(x)
        x = self.act(x)
        x = self.fc2(x)
        return x

# ResMLP blocks: a linear between patches + a MLP to process them independently
class ResMLP_BLocks(nn.Module):
    def __init__(self, nb_patches ,dim, layerscale_init):
        super().__init__()
        self.affine_1 = Affine(dim)
        self.affine_2 = Affine(dim)
        self.linear_patches = nn.Linear(nb_patches, nb_patches) #Linear layer on patches
        self.mlp_channels = Mlp(dim) #MLP on channels
        self.layerscale_1 = nn.Parameter(layerscale_init * torch.ones((dim))) #LayerScale
        self.layerscale_2 = nn.Parameter(layerscale_init * torch.ones((dim))) # parameters

    def forward(self, x):
        res_1 = self.linear_patches(self.affine_1(x).transpose(1,2)).transpose(1,2)
        x = x + self.layerscale_1 * res_1
        res_2 = self.mlp_channels(self.affine_2(x))
        x = x + self.layerscale_2 * res_2
        return x

# ResMLP model: Stacking the full network
class ResMLP_models(nn.Module):
    def __init__(self, dim, depth, nb_patches, layerscale_init, num_classes):
        super().__init__()
        self.patch_projector = Patch_projector()
        self.blocks = nn.ModuleList([
            ResMLP_BLocks(nb_patches ,dim, layerscale_init)
            for i in range(depth)])
        self.affine = Affine(dim)
        self.linear_classifier = nn.Linear(dim, num_classes)

    def forward(self, x):
        B, C, H, W = x.shape
        x = self.patch_projector(x)
        for blk in self.blocks:
            x = blk(x)
        x = self.affine(x)
        x = x.mean(dim=1).reshape(B,-1) #average pooling
        return self.linear_classifier(x)
```

---

## D   Additional Ablations

**Training recipe.**   DeiT [56] proposes a training strategy which allows for data-efficient vision transformers on ImageNet only. In Table D.1 we ablate each component of the DeiT training to go back to the initial ResNet50 training. As to be expected, the training used in the ResNet-50 paper [23] degrades the performance.

**Training schedule.**   Table D.2 compares the performance of ResMLP-S36 according to the number of training epochs. We observe a saturation of the performance after 800 epochs for ResMLP. This saturation is observed in DeiT from 400 epochs. So ResMLP needs more epochs to be optimal.

| Ablation | Imagenet1k-val top1-acc (%) |
|---|---|
| DeiT-style training | 76.6 |
| ─ CutMix [64] - Mixup [65] | 76.0 |
| ─ Random-erasing [67] | 75.9 |
| ─ RandAugment [13] | 73.2 |
| SGD optimizer | 72.1 |
| ─ Stochastic-depth [29] | 70.7 |
| ─ Repeated-augmentation [26] | 69.4 |
| 120 epochs | 67.7 |
| Step decay | 63.5 |
| Batch size 256 | 69.3 |
| ResNet-50 training 90 epochs | 69.2 |

Table D.1: Ablations on the training strategy

| Epochs | 300 | 400 | 500 | 800 | 1000 |
|---|---|---|---|---|---|
| Inet-val | 79.3 | 79.7 | 80.1 | **80.4** | 80.3 |
| Inet-real | 85.5 | 85.6 | **85.9** | 85.8 | 85.7 |
| Inet-V2 | 68.0 | 68.4 | 68.4 | 68.9 | **69.0** |

Table D.2: We compare the performance of ResMLP-S36 according to the number of training epochs.

**Pooling layers.** Table D.3 compare the performance of two pooling layers: average-pooling and class-MLP, with different depth with and without distillation. We can see that class-MLP performs much better than average pooling by changing only a few FLOPs and number of parameters. Nevertheless, the gap seems to decrease between the two approaches with deeper models.

| #layers | Params $\times 10^6$ | Flops $\times 10^9$ | Training | Pooling layer | top-1 acc. on ImageNet | | |
|---|---|---|---|---|---|---|---|
| | | | | | Inet-val | Inet-real | Inet-V2 |
| 12 | 15.4 | 3.0 | regular | average | 76.6 | 83.3 | 64.4 |
| 24 | 30.0 | 6.0 | regular | average | 79.4 | 85.3 | 67.9 |
| 36 | 44.7 | 8.9 | regular | average | 79.7 | 85.6 | 68.4 |
| 12 | 17.7 | 3.0 | regular | class-MLP | 77.5 | 84.0 | 66.1 |
| 24 | 32.4 | 6.0 | regular | class-MLP | 79.8 | 85.6 | 68.6 |
| 36 | 47.1 | 8.9 | regular | class-MLP | 80.5 | 86.3 | 69.5 |
| 12 | 15.4 | 3.0 | distillation | average | 77.8 | 84.6 | 66.0 |
| 24 | 30.0 | 6.0 | distillation | average | 80.8 | 86.6 | 69.8 |
| 36 | 44.7 | 8.9 | distillation | average | 81.0 | 86.8 | 70.2 |
| 12 | 17.7 | 3.0 | distillation | class-MLP | 78.6 | 85.2 | 67.3 |
| 24 | 32.4 | 6.0 | distillation | class-MLP | 81.1 | 87.0 | 70.4 |
| 36 | 47.1 | 8.9 | distillation | class-MLP | 81.4 | 87.3 | 70.7 |

Table D.3: We compare the performance of two pooling layers: average-pooling and class-MLP, with ResMLP-S architecture. We compare different depth, regular training and distillation.