# OpenReview forum: "ResMLP: Feedforward networks for image classification with data-efficient training"
_NeurIPS.cc/2021/Conference — NeurIPS 2021 Submitted_

### Official Review · Reviewer_Q86e · 2021-07-15

**Rating:** 7
**Confidence:** 4

**Summary:**

This paper introduce the ResMLP architecture. An alternative to self-attention layer in vision transformer using simply a $T \times T$ linear layer to allow long-range communication between tokens. They show that the performance of this model is comparable to the original vision transformer in supervised, self-supervised and transfer learning as well as knowledge distillation.

**Main Review:**


I would like to thank the authors for this interesting piece of work.
The proposed model is simple and well motivated. The extended ablation studies give a good insight of what makes the success of different token mixing methods.
The empirical performance is competitive with Vision Transformers.
The main limitation of the proposed architecture with linear token communication is that it operates on fixed size input images, contrary to convolutional or attention networks combined with a global pooling strategy.
Also it cannot learn data-dependant long range dependancies, which is verified by the similar performance of conv 3x3.
I would expect ResMLP to fail on the Long Range Arena [1] but I am curious of the results, maybe you could mention it in future work.

**Concerning communication accross patches.**
The extensive ablation studies are very insightful. The comparison with concurent work using MLP mixing is interesting.
MLP communication improves performance compared to Linear layer with only a minor increase in number of parameter, but with a significant increase in FLOPS.
The most insightful discussion is on comparing with conv 3x3: in most aspects convolution is better than linear mixing.

**Concerning self-supervised learning with DINO.**
One of the limitation of the linear communication of ResMLP compared to multi-head self-attention in Vision Transformers is that there is no attention maps per head to extract segmentation maps. The data independent ````''single head'' linear mixing would not exhibit object-centric behavior similar to Figure 3 in DINO.

**Application to Neural Machine Translation.**
It seems that you did not apply ResMLP to cross-attention, is there a fundamental limitation that could not allow it?
How do you handle masking in the decoder? Are you simply masking the weights with a triangular matrix?



Have you tried to process upscaled images at resolution 384x384 as it is commonly done with vision transformer?
The proposed method of only having a linear mixing layer gives a principle way to interpolate the weights of the matrix compared to concurrent patch communication approaches such as MLP-mixer. It might not even require finetuning.

Remarks:
- Figure 3: can you show the sparsity metric at initialization to have an horizontal bar as a reference?
- Missing related work [2] on early application of vision transformer. They showed that the learned attention maps were sparse and localised similarly to convolution which complement your finding on ResMLP.

[1] Long Range Arena: A Benchmark for Efficient Transformers
Yi Tay, Mostafa Dehghani, Samira Abnar, Yikang Shen, Dara Bahri, Philip Pham, Jinfeng Rao, Liu Yang, Sebastian Ruder, Donald Metzler
abs/2011.04006

[2] On the Relationship between Self-Attention and Convolutional Layers
Jean-Baptiste Cordonnier, Andreas Loukas, Martin Jaggi
ICLR 2020

**Time Spent Reviewing:**

7

---

> ### Author Response · Authors · 2021-08-10
> **Response to Reviewer Q86e**
>
> We thank the reviewer for his constructive feedback and as well as for all the suggestions to improve the paper.
>
> - *“Application to Neural Machine Translation. It seems that you did not apply ResMLP to cross-attention, is there a fundamental limitation that could not allow it? How do you handle masking in the decoder? Are you simply masking the weights with a triangular matrix?”*
>
> It is possible to use the same approach to substitute the cross-attention in the decoder, doing something similar to our approach used in appendix A to adapt class-attention for MLP.
>
> Yes, in the decoder, we are indeed simply masking the weights to enforce auto-regressive decoding.
>
> - *“I would expect ResMLP to fail on the Long Range Arena [1] but I am curious of the results, maybe you could mention it in future work.”*
>
>   *“Have you tried to process upscaled images at resolution 384x384 as it is commonly done with vision transformer? The proposed method of only having a linear mixing layer gives a principle way to interpolate the weights of the matrix compared to concurrent patch communication approaches such as MLP-mixer. It might not even require finetuning.”*
>
> Thanks for the suggestions, we will try it out.
>
> - *“Figure 3: can you show the sparsity metric at initialization to have an horizontal bar as a reference?”*
>
> Yes, we will add this. Thanks for the suggestion.
>
> - *“Missing related work [2] on early application of vision transformer. They showed that the learned attention maps were sparse and localised similarly to convolution which complement your finding on ResMLP.”*
>
> We will add these references in our related work.
>
> We hope this answers the reviewers' questions. We will be happy to answer any further questions.

---

> > ### Comment · Reviewer_Q86e · 2021-08-25
> > **Reviewer Q86e' answer**
> >
> > I would like to thank the authors for their answers. I read all the reviews and their associated answers.
> >
> > All reviewers seem to agree that the paper is well written and the experiments are insightful. I would not reject this paper based on results that do not reach the state of the art. The remaining question is if the study of the res-MLP architecture has significant novelty. Even though it might not have much impact for practitioners, the simplicity of this architecture (only matrix multiplications and non-linearities) could lead to interesting theoretical contributions on an architecture that exhibits good performance on images. I will keep my score unchanged.

---

### Official Review · Reviewer_rpHJ · 2021-07-16

**Rating:** 6
**Confidence:** 4

**Summary:**

Recently, vision Transformers have been popular and achieve SOTA performance on various tasks. This paper proposes a more simple variant of the Transformer architecture, *i.e.*, connect image patches via simple linear/MLP layers. The proposed ResMLP shows promising performance under various setups, *e.g.*, supervised and self-supervised training, and on different datasets. It is a new exploration of the methodology of neural architectures.

**Limitations And Societal Impact:**

As the authors indicate, ResMLP is only studied on image classification. Explorations on downstream tasks are expected if possible.
The main concern lies in the real value MLP carries, which is suggested to be demonstrated in rebuttal.

**Main Review:**

**Originality:**
Using MLP-like modules to construct the Transformer network, instead of self-attention, is novel. Though this idea is also proposed in concurrent works, this paper provides some new insights about MLP-like architectures, e.g., the affine transformation module to help training.

**Quality:**
The empirical studies are adequate. The experiments are performed on various datasets and setups. It is suggested to provide some theoretical insights if possible.

**Clarity:**
The submission is written clearly and easy to follow.

**Significance:**
The MLP-like architecture is a novel variant of Transformer architectures, which may bring some heuristic ideas for future explorations about this field.

Here are some questions and issues to be answered or resolved.
1. Though ResMLP shows promising performance in various experiments, the MLP-like architecture still does not show an evident advantage over attention/convolution-based ones, *e.g.*, ResMLP-S24 shows a lower accuracy than DeiT-S with worse throughput and FLOPs. I admit that the MLP architecture is a novel attempt at the Transformer structure. However, I still want to hear about the real advantage/potential MLP carries.
2. The linear-based patch interacting manner first transposes the matrix and then applies the linear projection, which equals a depthwise convolution with a global receptive field and shared parameters over channels. So what is the intrinsic difference between the MLP design and convolution? And it is more reasonable to compare ResMLP with a global receptive field (14x14) depthwise convolution in Tab. 3. In addition, the depthwise convolution in Tab. 3 shows similar accuracy with a smaller budget, *e.g.*, Params, and FLOPs, which makes the advantage of the MLP architecture confusing.
3. Some typos need to be checked and resolved, *e.g.*, "a convolutional layers" in line 90.

**Time Spent Reviewing:**

3.5 hours

---

> ### Author Response · Authors · 2021-08-10
> **Response to Reviewer rpHJ**
>
> We thank the reviewer for the constructive feedback.
>
> - *1) Though ResMLP shows promising performance in various experiments, the MLP-like architecture still does not show an evident advantage over attention/convolution-based ones, e.g., ResMLP-S24 shows a lower accuracy than DeiT-S with worse throughput and FLOPs. I admit that the MLP architecture is a novel attempt at the Transformer structure. However, I still want to hear about the real advantage/potential MLP carries.*
>
> Our ResMLP is the first to achieve competitive performance on complex tasks such as ImageNet classification with a full MLP-like architecture. Indeed, our image classification results on ImageNet-1k only are better than ViT in the paper of Dosovitskiy et al. [1] . We have benefited from the training improvements of DeiT  [2] (as discussed in Appendix D)  but still ResMLPs are probably far from being trained in an optimal way. Since this architecture has different properties than the one used until now. See also our answer to other reviewers.
>
> ResMLP, whose residual blocks consist of a one-hidden layer feed-forward network and a linear patch interaction layer (See Figure 1) is simpler and more stable architecture than ViT.  In our MLP architecture there is no self-attention and the stability problems that come with it (as shown in Liu et al.[3]) and no normalization. While ViT does not work without normalization.
>
> Compared to convnet our architecture is also simpler: We don't have an architecture with downsampling and we can easily remove the normalization while it is not so simple for convnet as shown in the NFNets paper [4]. Moreover, ResMLPs allow long range interaction (See Figure 2). The architecture has less priors than convnet and could help to better understand the latters.
>
> In summary, the purpose of the ResMLP paper is not to propose a new state of the art architecture but to show that a simple MLP has a good performance on different complex tasks such as image classification and machine translation, which has not been established yet.
> We have tried to have the simplest architecture possible as shown in our ablation Table 3. We choose simplicity over performance, and provide ablations to show that more complex variants can perform slightly better.
>
> [1] Dosovitskiy et al., An Image is Worth 16x16 Words: Transformers for Image Recognition at Scale, ICLR 2021
>
> [2] Touvron et al., Training data-efficient image transformers & distillation through attention, ICML 2021
>
> [3] Liu et al.,Understanding the Difficulty of Training Transformers, EMNLP 2020
>
> [4] Brock et al., High-Performance Large-Scale Image Recognition Without Normalization, ICML 2021
>
> - *2) The linear-based patch interacting manner first transposes the matrix and then applies the linear projection, which equals a depthwise convolution with a global receptive field and shared parameters over channels. So what is the intrinsic difference between the MLP design and convolution? And it is more reasonable to compare ResMLP with a global receptive field (14x14) depthwise convolution in Tab. 3. In addition, the depthwise convolution in Tab. 3 shows similar accuracy with a smaller budget, e.g., Params, and FLOPs, which makes the advantage of the MLP architecture confusing.*
>
> We agree that it is possible to present and implement a linear layer as a convolution (e.g. with the convolution module of any python library using a kernel of the size of the image and a stride of 0). Nevertheless, with a receptive field that remains the size of the image throughout the whole architecture, there is no real point in presenting it as a convolution mask.
>
> Indeed, in Table 3 the depthwise convolution shows similar accuracy with a smaller budget; this is an expected result of the MLPs being somehow over-parameterized in the first layers, it will converge to a convolution (See Figure 2). So MLP are very sparse in the first layers and can benefit from quantization to reduce parameters and FLOPs.
>
> Nevertheless, the purpose of the paper is not to be state-of-the-art for the parameters or FLOPs accuracy trade-off. But to show that with an extremely simple architecture such as an MLP it is possible to have good performance on many tasks while the MLP can appear as an architecture too simple to perform well with complex tasks.
>
> - *3) Some typos need to be checked and resolved, e.g., "a convolutional layers" in line 90.*
>
> Thank you for pointing them out, we will correct them.
>
> - *“As the authors indicate, ResMLP is only studied on image classification. Explorations on downstream tasks are expected if possible”*
>
> In Section 3.6 we provide Machine translation experiments.
>
> We hope this answers the reviewers' questions. We will be happy to answer any further questions.

---

> > ### Comment · Reviewer_rpHJ · 2021-08-25
> > **Thanks for Your Response**
> >
> > I sincerely appreciate the authors' feedback. However, my concern about the value MLP-like architectures carry has not been cleared yet. I agree with the authors' opinion that this paper is not pursuing SOTA results but proving such a simple architecture can work well. Nevertheless, beyond the MLP architecture can work well, what value can ResMLP bring? Since there is still a performance gap between MLP and CNN/Transformers, most researchers may still choose to perform works on CNN/Transformers. The claimed simpler architecture design seems not so significant.
> >
> > -------
> > Again, thanks for the detailed response. Considering the real value MLP-like architectures can bring, I decided to maintain my rating.

---

### Official Review · Reviewer_NGeM · 2021-07-18

**Rating:** 5
**Confidence:** 3

**Summary:**

This paper introduces Residual Multi-Layer Perceptrons (ResMLP): a purely multi-layer perceptron (MLP) based architecture for image classification. ResMLP takes image patches as input, projects them with a linear layer, and sequentially updates them in turn with two residual operations: (i) a simple linear layer that provides interaction between the patches, which is applied to all channels independently; and (ii) an MLP with a single hidden layer, which is independently applied to all patches. At the end of the network, the patches are average pooled, and fed to a linear classifier. The main change from ViT to ResMLP is replacing the self-attention layer with a simple linear layer cross patches. There are some other small changes, like pooling layers and normalization layers.

The claimed contributions are:
1. despite their simplicity, Residual Multi-Layer Perceptrons reach surprisingly good accuracy/complexity trade-offs with ImageNet-1k training only. From this point of view, this "surprisingly good" is very subjective. At least, from the results, the accuracy/complexity trade-offs is not as good as Vision Transformers (DeiT models).
2. The ResMLP benefits from distillation methods and also works well with self-supervised method DINO.
3. The simple linear layer enables observations on the spatial interaction that the network learns across layers.
4. The authors also try ResMLP on machine translation.



**Ethical Concerns:**

No.

**Limitations And Societal Impact:**

Yes.

**Main Review:**

This paper provides a comprehensive study on ResMLP, which aims to replace self-attention layers in ViT with MLPs. There are lots of experiments (trained on ImageNet1K only) to show the effect of different designs and training recipes. It is a good literature for researchers who are interested in this area but do not have rich resources to conduct such extensive studies.

However, from my understanding, the results in fact seem to be negative for ResMLP, when compared with ViT. Therefore, I doubt that the main contribution of this paper, "ResMLP reaches surprisingly good accuracy/complexity trade-offs with ImageNet-1k training only", is valid or not in this paper. In Table 1, when we examine the trade-off between Top-1 acc and throughput, the Transformer networks clearly win over the feedforward networks. Moreover, because linear layers are highly optimized in Pytorch, there should be no excuse from the implementation of ResMLP. Compared with ResNets, both ViTs and ResMLPs removed the strong inductive bias, e.g., locality and translational invariance. Although ResMLP (with a linear layer) sounds simpler than ViT (with a self-attention layer), their cross patch interactions are equally general and I do not think that ResMLP use less inductive bias than ViT. From my understanding, "minimal inductive bias" and "good accuracy/complexity trade-offs" are the main claimed advantage of ResMLP. But I do not see its advantage over its exact baseline, ViT. Do we have a positive result for ResMLP compared with ViT counterparts?

Transformers has shown to be very successful in the claimed 2/3/4 contributions. We can only see that ResMLP can also do 2/3/4, but in fact, from the current results, seems that transformers are even doing better in terms of the 2/3/4 contributions.

Detailed comments:
1. The comparison between "ResMLP-S12 with linear -> conv 3x3 depth-wise" and its baseline "ResMLP-S12" in Table 3. Why there is only little parameter saving compared with the baseline (15.4 ->14.9)? Is it because that the main number of parameters is in the channel-wise MLP layers? If so, it will be helpful for the authors to provide the comparison of number of parameters in patch-wise interaction layers and the channel-wise MLP layers.
2. Line 217: "differential" should be difference.
3. In Table 4, ResMLP performs clearly worse than ResNets on the last three datasets (cars/iNat18/iNat19). The first three sets (Cifar10, Cifar100, Flowers102) have very small domain gap with the pretrained ImageNet1k dataset, but the last three are expected to have larger domain gap. Does this table show that ResMLP is worse than ResNets in terms of generalization?
Line 267-268: "We initialize the LayerScale parameter to 0.2" I guess that the authors do a hyperparameter search here. Then does this mean that the default small initialization of LayerScale does not work here? Why?

In Appendix A:
Class-attention!class-MLP: "If we further replace the class-attention layer of CaiT by a MLP as
described in our paper, then we obtain an attention-free network whose top-1 accuracy on Imagenet-val is 79.2%". Which model is this 79.2% one in the expansion factor table? The x0.5 one, or the x1 one?

Ablation on the size of the communication MLP: "For the MLP that replaced the class-attention, we have explored different sizes of the latent layer, by adjusting the expansion factor e ... This has motivated the choice of adopting a simple linear layer of size N ×N: This subsequently improved performance to 79.5% in a
setting comparable to the table above". The ablation study of expansion factor e is on the class-attention layers here, but then it motivated the choice of adopting a simple linear layer in the layers that replacing self-attention with two-layer MLPs? Have the authors tried to search for a better expansion factor e for two-layer MLPs in the patch-wise interaction layers, i.e., the "Patch communication: linear->MLP" in Table 3? In Table 3, it seems that "Patch communication: linear->MLP" performs better than the final simple linear choice. This conflicts with the claim here that simple linear layer improves the performance... Can the authors provide a concrete comparison with the two-layer MLP and a simple linear layer for replacing self-attention layers?

In Appendix D
Table D.1: This table is wrong? "ResNet-50 training 90 epochs" should be around 76.5 (in stead of the reported 69.2) and the "DeiT-style training" should be much higher than the reported number 76.6.

**Time Spent Reviewing:**

5

---

> ### Author Response · Authors · 2021-08-10
> **Response to Reviewer NGeM**
>
> We thank the reviewer for the constructive feedback.
>
> - *“this "surprisingly good" is very subjective. At least, from the results, the accuracy/complexity trade-offs is not as good as Vision Transformers (DeiT models).”*
>
>      *“However, from my understanding, the results in fact seem to be negative for ResMLP, when compared with ViT. Therefore, I doubt that the main contribution of this paper, "ResMLP reaches surprisingly good accuracy/complexity trade-offs with ImageNet-1k training only", is valid or not in this paper. In Table 1, when we examine the trade-off between Top-1 acc and throughput, the Transformer networks clearly win over the feedforward networks.”*
>
>     *“Do we have a positive result for ResMLP compared with ViT counterparts?”*
>
> Our paper is a proof of concept that a simple MLP architecture performs quite well while being easy to train on complex tasks such as image classification or machine translation.
> We do not claim to propose a new state of the art architecture: We chose simplicity over performance. We selected the simplest architecture possible as shown in our ablation Table 3.
>
> ResMLP, whose residual blocks consist of a one-hidden layer feed-forward network and a linear patch interaction layer is simpler and more stable architecture than ViT.  In our MLP architecture there is no self-attention and the stability problems that come with it (as shown in Liu et al. [3]) and no normalization (ViT, like other networks, do not work without normalization).
>
> Note that our image classification results on ImageNet-1k only are better than the original ViT by Dosovitskiy et al.[1]. We have benefited from the training improvements of DeiT [2] (as discussed in Appendix D), but this training remains suboptimal for ResMLP (see details in answers to other reviewers). We have adopted existing training strategies to avoid changing both the architecture and the training method.
>
>
> [1] Dosovitskiy et al., An Image is Worth 16x16 Words: Transformers for Image Recognition at Scale, ICLR 2021
>
> [2] Touvron et al., Training data-efficient image transformers & distillation through attention, ICML 2021
>
> [3] Liu et al.,Understanding the Difficulty of Training Transformers, EMNLP 2020
>
> - *"The comparison between "ResMLP-S12 with linear -> conv 3x3 depth-wise" and its baseline "ResMLP-S12" in Table 3. Why there is only little parameter saving compared with the baseline (15.4 ->14.9)? Is it because that the main number of parameters is in the channel-wise MLP layers? If so, it will be helpful for the authors to provide the comparison of number of parameters in patch-wise interaction layers and the channel-wise MLP layers."*
>
> Yes, the main number of parameters is in the channel-wise MLP layers.
> As explained, in equation 2 and 3 with  “The dimensions of the parameter matrix A are N2×N2” L82 and “ Finally, the weight matrices B and C have the same dimensions as in a Transformer layer, which are 4d×d and d×4d, respectively” L85.
> For the model small at resolution 224 we have N = 14 and d = 384. So for one block the patch-wise interaction layers have 38,416 parameters and the channel-wise MLP layers have 1,179,648 parameters.
>
> - *"Line 217: "differential" should be difference."*
>
> Thanks for pointing this out, we will correct the typo.
>
> - *“In Table 4, ResMLP performs clearly worse than ResNets on the last three datasets (cars/iNat18/iNat19). The first three sets (Cifar10, Cifar100, Flowers102) have very small domain gap with the pretrained ImageNet1k dataset, but the last three are expected to have larger domain gap. Does this table show that ResMLP is worse than ResNets in terms of generalization? “*
>
> Excellent remark, so yes but this is likely because the training of MLP is less adapted than that of the ResNet-50 models in Table 4, as discussed hereafter. The ResNet-50 were trained and fine-tuned with a strong procedure from the Grafit paper [1] (They reach 79.3% on ImageNet). For ResMLP we used the DeiT finetuning procedure in order to have a more direct comparison with ViT. It is suboptimal for MLPs.
>
> Now if we slightly adjust the hyperparameters to ResMLP, the performance increases significantly and outperforms the competitive ResNet-50 from the Grafit paper on Cars. We observe interesting gains on iNaturalist. More precisely: with ResMLP-S24 we reach 90.2% on Cars-196, 65.4% on INaturalist 2018 and 73.0% on iNaturalist 2019. These improvements should certainly be pushed further as we have not put much emphasis on pushing these numbers.
> In summary, with a few adjustments of hyper-parameters, ResMLP are similar to ResNet in terms of generalization except on iNaturalist. We will discuss this in the final version.
>
> [1]  Touvron et al., Grafit: Learning fine-grained image representations with coarse labels, ICCV 2021
>
> - *"We initialize the LayerScale parameter to 0.2" I guess that the authors do a hyperparameter search here. Then does this mean that the default small initialization of LayerScale does not work here? Why?”*
>
> We haven't done a grid search, we followed the recommendations of the layer-scale paper [1] for all your experiments. The point mentioned here is only for the NLP task with the 6 layers model. In the layer scale paper they use 0.1 for 12 layers but they don’t experiment with 6 layers models. We have therefore scaled this parameter proportionally.
>
> [1] Touvron et al., Going deeper with Image Transformers, ICCV 2021
>
> - *"In Appendix A: Class-attention!class-MLP: "If we further replace the class-attention layer of CaiT by a MLP as described in our paper, then we obtain an attention-free network whose top-1 accuracy on Imagenet-val is 79.2%". Which model is this 79.2% one in the expansion factor table? The x0.5 one, or the x1 one? The ablation study of expansion factor e is on the class-attention layers here, but then it motivated the choice of adopting a simple linear layer in the layers that replacing self-attention with two-layer MLPs?"*
>
> Our apologies, there is a typo in appendix A in the paragraph: “Ablation on the size of the communication MLP”. The sentence:  “For the MLP that replaced the **class-attention**,”  should be “For the MLP that replaced the **self-attention**,”.We will correct it.
>
> In Appendix A, we have evaluated each of the changes independently unless mentioned otherwise.
>
> - *"In Table 3, it seems that "Patch communication: linear->MLP" performs better than the final simple linear choice. This conflicts with the claim here that simple linear layer improves the performance... Can the authors provide a concrete comparison with the two-layer MLP and a simple linear layer for replacing self-attention layers?"*
>
> In Table 3 the default model is without normalization, in Appendix A the default model is with LayerNorm. So in terms of  absolute performance, in a transformer architecture it is better to replace the two-layer MLP by a simple linear layer. But if you remove the normalization it is better to have a two-layer MLP.
>
> With and without normalization the model with two-layer MLP has more FLOPs and parameters than the model with a simple linear layer (see Table 3). So we can't compare them directly. We will complete the table with other models for comparison points for the trade-off FLOPs accuracy for two-layer MLP vs linear layer.
>
> That being said, our objective was to favor simplicity over the best performance. For simplicity and interpretability reasons we chose to use only a linear layer, (see Figure 2 and B.1) without a big gap in terms of accuracy/flops trade-off. Our ablations already provide different variants with better performance for the readers interested in pushing numbers.
>
> - *"In Appendix D Table D.1: This table is wrong? "ResNet-50 training 90 epochs" should be around 76.5 (instead of the reported 69.2) and the "DeiT-style training" should be much higher than the reported number 76.6"*
>
> As explained,  Appendix D paragraph “Training recipe.”   ``In Table D.1 we ablate each component of the DeiT training to go back to the initial ResNet50 training.” so the row “ResNet-50 training 90 epochs'' in  Table D.1 correspond to ResMLP-S-12 trained with the “ResNet-50 training 90 epochs'' training strategy (random resized crop, color jitter, step decay learning rate). We will clarify this in the caption.
>
> We hope this answers the reviewers' questions. We will be happy to answer any further questions.

---

> > ### Comment · Reviewer_NGeM · 2021-09-08
> > **Thanks for Your Response**
> >
> > Thank the authors for the detailed replies! It resolves most of my confusions. For the typo/confusion in the appendix, it remains. The authors may make the appendix more clearer in the revision.
> >
> > The main advantage of ResMLP is its conceptual simplicity. This paper did a thorough experimental study for this conceptually simpler architecture. However, after reading the replies and other reviewers' comments, I think that my main concern for ResMLP still exists: when compared with the ViT architecture, the ResMLP does not show much advantage, even a clear disadvantage in terms of accuracy/complexity trade-off.
> >
> > I would like to keep my rating.

---

### Official Review · Reviewer_x4AE · 2021-08-03

**Rating:** 5
**Confidence:** 4

**Summary:**

The paper proposes an MLP-based architecture for sequence-based DL tasks (vision, NLP). Basically, the idea is to remove the self-attention primitive from Transformers. Instead, it transposes the seq and channel dimension and applies linear projections. This is the same idea as the concurrent work MLP-Mixer from Google. The spatial mixing is now basically a DWise Conv with full receptive field shared across channels. Paper shows reasonable results on ImageNet benchmark (needs a lot more params and worse on throughput for the same top-1 accuracy wrt ViT, and also worse than pure ConvNet baselines). Paper has plenty of ablations on the training recipe and architecture. NLP tasks only include short sequence benchmarks like Translation, with promising signs.

**Main Review:**

Pros:
1. Timely paper - along with several concurrent works - on exploring MLPs for sequence models
2. Removing LayerNorm and CLS tokens
3. Benchmarking on multiple tasks and modalities
4. Good ablation studies

Cons:
1. Lukewarm results (not the fault of authors, but it is super clear self-attention is absolutely necessary to get better numbers)
2. Paper sells the idea of FCNs a lot - but it's pretty clear the architecture is a reparametrized ConvNet in disguise - locality and sparsity are what make it work well and scale well
3. Should show more results on a different vision task - ex detection/segmentation where pairwise interactions are known to help/matter more
4. No results on BERT benchmarks would be nice to have.
5. Studies on pre-training with more data.

**Time Spent Reviewing:**

2

---

> ### Author Response · Authors · 2021-08-10
> **Response to Reviewer x4AE**
>
> We thank the reviewer for the constructive feedback.
> -  *“This is the same idea as the concurrent work MLP-Mixer from Google.”*
>
> Our ResMLP as well as the concurrent MLP-Mixer are the first to demonstrate that a full MLP-like architecture can perform well on complex tasks such as ImageNet classification. We agree that the general idea is similar in these papers.
>
> We point out significant differences wrt MLP-Mixer: our ResMLP is simpler (Token mixing with only one linear layer, no normalization), more efficient (better performance on ImageNet without extra data, less FLOPs and parameters) and we cover machine translation tasks
> (MLP-Mixer only considers image classification).
>
> We also include analysis and ablations that are different from Google's paper (ablations of variants and of training paradigms like self-supervised pre-training and distillation), which we believe have some value for the community.
>
> - *"Lukewarm results (not the fault of authors, but it is super clear self-attention is absolutely necessary to get better numbers)"*
>
> Despite its simplicity, ResMLP, whose residual blocks consist of a one-hidden layer feed-forward network and a linear patch interaction layer, achieves an unexpectedly high performance on ImageNet and offers excellent accuracy/complexity trade-offs with ImageNet-1k training only (See our ablation Table 3). We agree that some existing architectures are better (and we include them in our comparison), but this is not our claim.
>
> - *“The spatial mixing is now basically a DWise Conv with full receptive field shared across channels.”* , *“Paper sells the idea of FCNs a lot - but it's pretty clear the architecture is a reparametrized ConvNet in disguise - locality and sparsity are what make it work well and scale well”*
>
> Figure 2 shows that the linear layer will sometimes converge to a kind of convolution (with non square kernel). Sometimes it converges to a pattern that provides longer-range interaction. When it happens, the interaction is global and not necessarily sparse (see Figure 2, layer 20).
>
> Table 3, we have some ablations with convolutions to better understand the role of the translation priors.
>
>
> - *“Should show more results on a different vision task - ex detection/segmentation where pairwise interactions are known to help/matter more” , “No results on BERT benchmarks would be nice to have.”*
>
> We thank the reviewer for their suggestion to add detection/segmentation experiments and BERT benchmarks. Currently, Sections 3.5 and 3.6 present some extensions on Transfer learning and Machine translation.
>
> - *“Studies on pre-training with more data.”*
>
> In Table 3, we include results with several pre-training, in particular we report an experiment with ImageNet-21k pre-training. We are not aware of a larger dataset publicly available.
>
> We hope this answers the reviewers' questions. We will be happy to answer any further questions.

---

### Decision · Program_Chairs · 2021-09-27

**Decision:**

Reject

**Comment:**

In summary, the reviewers see the paper as very close to the acceptance threshold with no reviewer opposing acceptance, but no reviewer strongly arguing for acceptance, either. Half of the reviewers are leaning towards accepting, half are leaning towards rejecting.

The paper discusses an MLP-based architecture for DL tasks, which results in surprisingly good performance on vision and NLP tasks.

Highlights of the paper mentioned by the reviewers include:
* This paper is concurrent (and partially complimentary) work with several other papers exploring a similar direction (as noted in footnote 1 and pointed out in reviews). This may indicate an interest in this line of work in the community.
* The paper is well-written and the model is well-motivated.

The main concerns expressed by the reviewers include the following:
* While the performance that is reached may be interesting, it is not clear whether the architecture can be relevant in practice when compared to ViTs or ResNets.
* Some related work could be discussed more appropriately.

Overall, the paper is very close to the acceptance threshold, but sufficient support for acceptance has not been reached.